# Shared and distinct genetic etiologies for different types of clonal hematopoiesis

Derek W. Brown [1,2,8], Liam D. Cato[3,4,8], Yajie Zhao [5,8], Satish K. Nandakumar [3,4,6], Erik L. Bao [3,4], Eugene J. Gardner[5], Aubrey K. Hubbard[1], Alexander DePaulis[1], Thomas Rehling[1], Lei Song[1], Kai Yu[1], Stephen J. Chanock [1], John R. B. Perry [5,7,9] ✉, Vijay G. Sankaran [3,4,9] ✉ & Mitchell J. Machiela [1,9] ✉

Clonal hematopoiesis (CH)—age-related expansion of mutated hematopoietic clones—can differ in frequency and cellular fitness by CH type (e.g., mutations in driver genes (CHIP), gains/losses and copy-neutral loss of chromosomal segments (mCAs), and loss of sex chromosomes). Co-occurring CH raises questions as to their origin, selection, and impact. We integrate sequence and genotype array data in up to 482,378 UK Biobank participants to demonstrate shared genetic architecture across CH types. Our analysis suggests a cellular evolutionary trade-off between different types of CH, with LOY occurring at lower rates in individuals carrying mutations in established CHIP genes. We observed co-occurrence of CHIP and mCAs with overlap at *TET2*, *DNMT3A*, and *JAK2*, in which CHIP precedes mCA acquisition. Furthermore, individuals carrying overlapping CH had high risk of future lymphoid and myeloid malignancies. Finally, we leverage shared genetic architecture of CH traits to identify 15 novel loci associated with leukemia risk.

Recent studies have reported the frequent occurrence and clonal expansion of post-zygotic mutations in the hematopoietic system, now seen in all human tissues but at different attained frequencies[1–6]. Initially, clonal expansion was recognized by the presence of skewed X chromosome inactivation[7,8]. Subsequent studies have revealed the presence of mosaic chromosomal alterations (mCAs), including frequent loss of X (LOX) and Y (LOY) chromosomes, in a subset of hematopoietic cells. Most recently, clonal expansion of recurrent somatic driver mutations observed in hematologic malignancies have been identified in individuals with otherwise normal hematologic parameters, a condition known as clonal hematopoiesis of indeterminate potential (CHIP). These somatic alterations can predispose to either myeloid or lymphoid malignancies, but do not necessarily progress; in other words, many otherwise healthy individuals are observed to have CH[9]. Moreover, recent studies have shown the additive impact of mCAs and CHIP mutations on predisposition to blood cancers, with respect to overall risk for a primary hematologic malignancy and also in the setting of therapy-associated myeloid malignancies[10,11], but do not systematically examine genetic relationships which can inform shared mechanisms.

[1]Division of Cancer Epidemiology and Genetics, National Cancer Institute, Rockville, MD, USA. [2]Cancer Prevention Fellowship Program, Division of Cancer Prevention, National Cancer Institute, Rockville, MD, USA. [3]Division of Hematology/Oncology, Boston Children's Hospital and Department of Pediatric Oncology, Dana-Farber Cancer Institute, Harvard Medical School, Boston, MA 02115, USA. [4]Broad Institute of MIT and Harvard, Cambridge, MA 02142, USA. [5]MRC Epidemiology Unit, Wellcome-MRC Institute of Metabolic Science, University of Cambridge School of Clinical Medicine, Cambridge CB2 0QQ, UK. [6]Department of Cell Biology, Albert Einstein College of Medicine, Albert Einstein Cancer Center, Ruth L. and David S. Gottesman Institute for Stem Cell Research and Regenerative Medicine, Bronx, NY 10461, USA. [7]Metabolic Research Laboratory, Wellcome-MRC Institute of Metabolic Science, University of Cambridge School of Clinical Medicine, Cambridge CB2 0QQ, UK. [8]These authors contributed equally: Derek W. Brown, Liam D. Cato, Yajie Zhao. [9]These authors jointly supervised this work: John R. B. Perry, Vijay G. Sankaran, Mitchell J. Machiela. ✉e-mail: john.perry@mrc-epid.cam.ac.uk; sankaran@broadinstitute.org; mitchell.machiela@nih.gov

Large studies have begun to reveal how germline genetic variants can increase risk for acquisition of CH but have yet to investigate germline factors leading to the co-existence of events or the biologic mechanisms of hematopoiesis that confer the risk of CH. Here, we perform a systematic investigation examining genetic and phenotypic associations across all types of CH utilizing a spectrum of genotyping and sequencing data from large-scale genetic susceptibility studies of types of clonal hematopoiesis (e.g., mCAs, LOX, LOY, and CHIP), hematologic malignancies (e.g., myeloproliferative neoplasms (MPNs)), and hematopoietic phenotypes. We identify numerous interrelated genetic and phenotypic associations between these distinct but potentially related phenotypes, providing substantial new insights into the shared and distinct mechanisms and consequences of different types of clonal hematopoietic expansions.

## Results

### CH states display shared genetic and phenotypic relationships

We began by investigating the co-existence of different types of CH: loss of chromosome Y (LOY) in men, loss of chromosome X (LOX) in women, autosomal mCAs including gains, losses, and copy neutral loss of heterozygosity (CNLOH), CHIP, and MPNs (Fig. 1). We subsequently examined associations with germline susceptibility variants, in anticipation of the discovery of shared elements. Genome-wide association study (GWAS) summary statistics for each type of CH were analyzed and pairwise genetic correlations between traits were computed (Online Methods). Using the high-definition likelihood (HDL) method, positive genetic correlations were observed between LOY and LOX ($\rho = 0.23$, $P = 5.53 \times 10^{-9}$), LOY and MPN ($\rho = 0.35$, $P = 1.74 \times 10^{-4}$), autosomal mCAs and MPN ($\rho = 0.57$, $P = 1.83 \times 10^{-3}$), and CHIP and MPN ($\rho = 0.48$, $P = 4.32 \times 10^{-3}$) (Fig. 2, Supplementary Data 1). We repeated genetic correlation analyses using linkage disequilibrium score regression (LDSC) (Online Methods) and likewise observed positive genetic correlations for LOY with both LOX ($\rho = 0.30$, $P = 4.09 \times 10^{-5}$) and MPN ($\rho = 0.21$, $P = 1.67 \times 10^{-2}$) (Supplementary Fig. 1, Supplementary Data 2). The genetic correlation between autosomal mCAs and MPN, and CHIP and MPN had the same direction of effect as found with HDL (Supplementary Fig. 1, Supplementary Data 2).

In an analysis of 482,378 subjects from the UK Biobank, we investigated adjusted phenotypic associations between types of CH (Online Methods, Supplementary Data 3). Consistent with previous studies of CH[12–14], each type of CH investigated demonstrated a strong positive association with age (Supplementary Fig. 2). We observed an inverse phenotypic association between LOY and MPN (T-statistic = $-4.76$, $P = 1.96 \times 10^{-6}$) (Fig. 3, Supplementary Data 4), which is opposite in direction from the genetic correlation. We were unable to evaluate the phenotypic association between LOY and LOX, as these are sex-specific traits.

We report positive phenotypic associations of autosomal mCAs with LOY (T-statistic = 4.31, $P = 1.61 \times 10^{-5}$), LOX (T-statistic = 12.44, $P = 1.54 \times 10^{-35}$), CHIP (T-statistic = 8.89, $P = 6.41 \times 10^{-19}$), and MPN (T-statistic = 41.00, $P < 5 \times 10^{-324}$) (Fig. 3, Supplementary Data 4). Sensitivity analyses removed individuals with mCAs spanning the *JAK2* region (N = 550), a region frequently impacted by mCAs in MPNs[15–18], and still observed a positive phenotypic association between autosomal mCAs and MPN, though the association was attenuated (T-statistic = 15.53, $P = 2.39 \times 10^{-54}$). CHIP was also positively associated with MPN (T-statistic = 8.82, $P = 1.18 \times 10^{-18}$) and inversely associated with LOY (T-statistic = $-4.11$, $P = 4.04 \times 10^{-5}$) (Fig. 3, Supplementary Data 4). The inverse association and exclusivity between CHIP and LOY were most prominent when stratified by the frequently observed CHIP gene mutations *DNMT3A* CHIP with LOY ($N_{CHIP} = 1{,}818$, T-statistic = $-3.71$, $P = 2.08 \times 10^{-4}$) and *TET2* CHIP with LOY ($N_{CHIP} = 786$, T-statistic = $-3.99$, $P = 6.50 \times 10^{-5}$) (Supplementary Data 5). Limited evidence was observed for an inverse relationship with LOY for less common CHIP mutations, suggesting *DNMT3A* and *TET2* mutations are primarily

responsible for the overall inverse relationship observed between CHIP and LOY. Sensitivity analyses restricting LOY to cell fractions greater than 15% to eliminate potential bias in detection differences between CHIP and mCAs provided some support of the inverse relationships between *DNMT3A* CHIP with LOY ($N_{CHIP} = 1467$, T-statistic = $-2.18$, $P = 2.91 \times 10^{-2}$) and *TET2* CHIP with LOY ($N_{CHIP} = 662$, T-statistic = $-1.71$, $P = 8.71 \times 10^{-2}$) (Supplementary Data 5). Associations by specific non-DNMT3A or non-TET2 CHIP genes displayed little evidence of directional relationship, perhaps due to small numbers of individuals with CHIP gene and LOY overlap. In further evaluation, we performed exome-wide burden analyses to identify rare (MAF < 0.1%) protein coding variants associated with LOY (Online Methods). These analyses identified three established CHIP genes at exome-wide significance (Supplementary Data 6), demonstrating that individuals carrying heterozygous loss-of-function variants in *TET2* ($n = 193$, beta = $-0.21$, SE = 0.03, $P = 7.7 \times 10^{-15}$), *ASXL1* ($n = 213$, beta = $-0.18$, SE = 0.03, $P = 1.3 \times 10^{-12}$), and *DNMT3A* ($n = 89$, beta = $-0.17$, SE = 0.04, $P = 2.2 \times 10^{-5}$) were less likely to exhibit LOY (Supplementary Figs. 3, 4, Supplementary Data 7). These findings reinforce the idea that acquiring LOY in the presence of some CHIP mutations is likely selected against in clonally-expanded hematopoietic stem cells.

We next examined the cellular fraction of individuals with autosomal mCA events and the variant allele frequency (VAF) of individuals with CHIP mutations and observed that individuals with higher cellular fractions of autosomal mCA events (i.e., greater proportion of cells carrying the somatic event) were more likely to have LOX (T-statistic= 3.08, $P = 2.09 \times 10^{-3}$), CHIP (T-statistic = 2.35, $P = 1.88 \times 10^{-2}$), and MPN (T-statistic = 17.83, $P = 3.17 \times 10^{-70}$) (Supplementary Fig. 5, Supplementary Data 8). Higher autosomal mCA cellular fraction was inversely associated with LOY (T-statistic = $-7.76$, $P = 9.77 \times 10^{-15}$) (Supplementary Fig. 5, Supplementary Data 8). Individuals with higher VAF of CHIP mutations (i.e., higher clonal fractions) were more likely to have detectable autosomal mCAs (T-statistic = 5.82, $P = 6.21 \times 10^{-9}$) and MPNs (T-statistic = 7.19, $P = 7.36 \times 10^{-13}$), and less likely to have LOY (T-statistic = $-3.80$, $P = 1.48 \times 10^{-4}$) (Supplementary Fig. 6, Supplementary Data 9).

In an analysis of co-existence of types of CH, CHIP and autosomal mCAs significantly co-occurred in the same individual (hypergeometric $P = 5.32 \times 10^{-28}$; Supplementary Fig. 7a) with 439 individuals (6.0% of individuals with CHIP, 6.3% of individuals with autosomal mCAs) carrying both (Supplementary Data 3). Individuals with autosomal mCAs displayed a distinct pattern of CHIP gene mutations compared to individuals without autosomal mCAs (Supplementary Fig. 7b, Supplementary Data 10). 13 CHIP gene mutations were significantly enriched in individuals with autosomal mCAs (*DNMT3A*, *TET2*, *ASXL1*, *TP53*, *SF3B1*, *STAT3*, *SRSF2*, *MPL*, *KRAS*, *JAK2*, *IDH1*, *PRPF40B*, and *PIGA*; Supplementary Fig. 7b), a similar pattern of co-occurrence as previously observed[10,11]. Additionally, individuals with CHIP mutations were more likely to acquire autosomal mCAs across 16 chromosomes (Supplementary Fig. 7c, Supplementary Data 11), with enrichment for several chromosome-specific copy number states (e.g., CNLOH in chromosomes 1, 4, and 9; Supplementary Fig. 7c, Supplementary Data 11).

An evaluation of the 439 individuals with both a CHIP mutation and autosomal mCA revealed that 53 (12.1%) had events spanning the same genomic region (binomial $P = 1.70 \times 10^{-10}$). 9 CHIP genes overlapped with autosomal mCAs, with *TET2* mutations accounting for 34 (54.0%) of the observed overlapping mutations (Supplementary Fig. 8). CNLOH was the most frequently observed autosomal mCA event ($N = 46$ (73.0%)) among all overlapping mutations (Supplementary Fig. 8). We examined the clonal fractions of both somatic mutations to provide a window into the clonal evolution of CHIP mutations and autosomal mCAs and found higher estimated CHIP VAF than estimated mCA cellular fraction in a majority of co-localizing mutations, suggesting the acquisition of the CHIP mutation preceded the acquisition

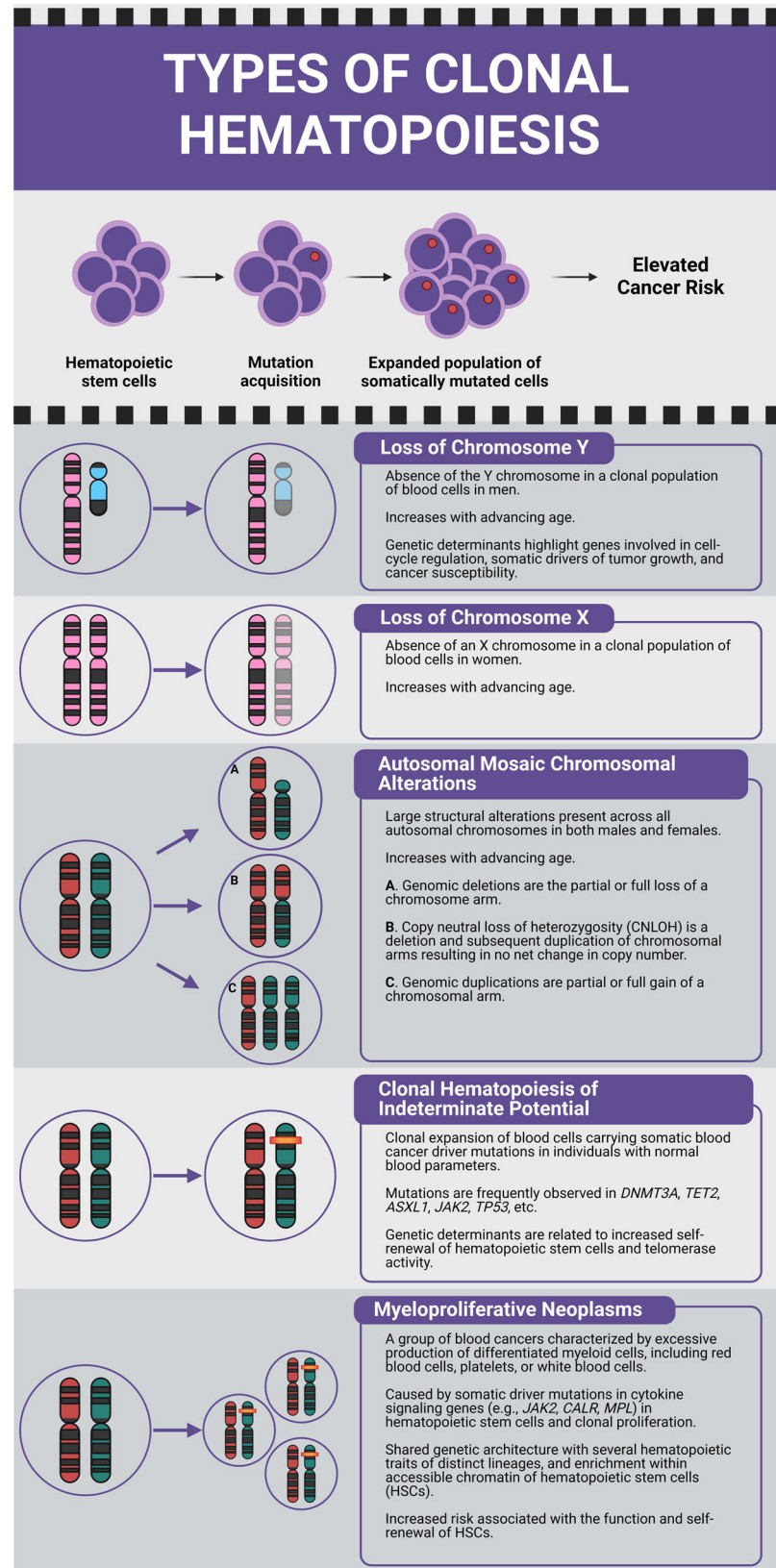

**Fig. 1 | Description of each type of clonal hematopoiesis (CH).** The top panel depicts the acquisition and clonal expansion of a CH clone and its potential for elevated hematologic cancer risk. Lower panels illustrate loss of chromosome Y (LOY); loss of chromosome X (LOX); autosomal mosaic chromosomal alterations (mCAs) that include (A) genomic deletions, (B) copy neutral loss of heterozygosity (CNLOH), and (C) genomic duplications; clonal hematopoiesis of indeterminate potential (CHIP); and myeloproliferative neoplasms (MPN).

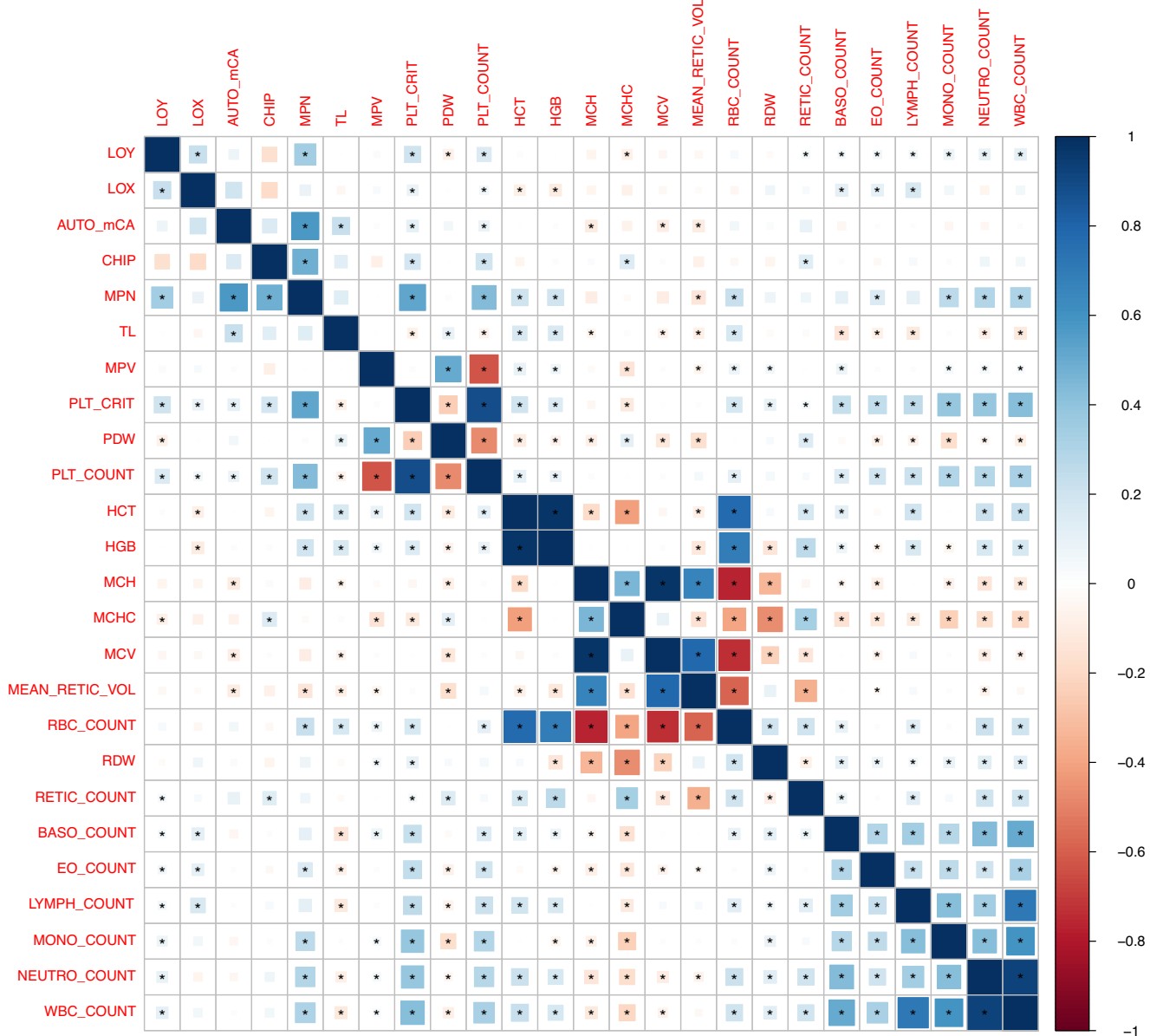

**Fig. 2 | Pairwise genetic correlations between each type of clonal hematopoiesis, telomere length, and 19 blood cell traits derived using the high-definition likelihood (HDL) method.** Square areas represent the absolute value of genetic correlations. Blue, positive genetic correlation; red, negative genetic correlation.

Genetic correlations that are significantly different from zero (*p*-value < 0.05) are marked with an asterisk. All pairwise genetic correlations and *p*-values are given in Supplementary Data 1.

of autosomal mCAs (binomial $P = 1.75 \times 10^{-4}$; Supplementary Fig. 9); this finding is consistent with a multi-hit hypothesis in driving clonal evolution. This is particularly evident in loss-of-heterozygosity of chromosome 9 alterations after acquisition of a *JAK2 V617F* mutation, as has been seen in individuals with MPNs[19–21]. Subsequent autosomal mCA-induced loss of heterozygosity or amplification of CHIP driver mutations could confer strong selective advantages promoting rapid cellular expansion. To test this hypothesis, we investigated the association between CHIP VAF and co-occurrence and overlap of CHIP and autosomal mCAs (Supplementary Fig. 10, Supplementary Data 12). We found that individuals with both CHIP and autosomal mCAs had higher CHIP VAF (T-statistic = 3.88, $P = 1.03 \times 10^{-4}$) and that individuals carrying overlapping CHIP and autosomal mCAs displayed further elevated CHIP VAF (T-statistic = 6.54, $P = 6.75 \times 10^{-11}$) compared to individuals with only CHIP mutations, demonstrating that individuals with CHIP and autosomal mCAs, especially those with overlapping mutations, have increased clonal expansion. As detection of CHIP

requires higher VAF than cell fractions required to detect autosomal mCAs, we performed sensitivity analyses restricting to autosomal mCAs with cell fractions similar to the detection level for CHIP mutations (cell fraction >15%) to eliminate any potential bias due to detection differences. Results from the sensitivity analysis showed similar significant associations with higher estimated effect sizes, further supporting potential mutational cooperativity between CHIP and autosomal mCAs (Supplementary Fig. 10, Supplementary Data 12).

We further investigated lymphoid and myeloid risk in individuals with and without CHIP and autosomal mCAs. Individuals with both CHIP and non-overlapping autosomal mCAs ($N = 386$) demonstrated a strong positive association with both incident lymphoid malignancy risk (hazard ratio (HR) = 8.63, 95% confidence interval (CI) = 5.93–12.58, $P = 3.09 \times 10^{-29}$) and incident myeloid malignancy risk (HR = 24.70, 95% CI = 14.82–41.16, $P = 7.98 \times 10^{-35}$) compared to individuals without CHIP or autosomal mCAs (Supplementary Fig. 11). Individuals carrying overlapping CHIP and autosomal mCAs ($N = 53$) displayed even

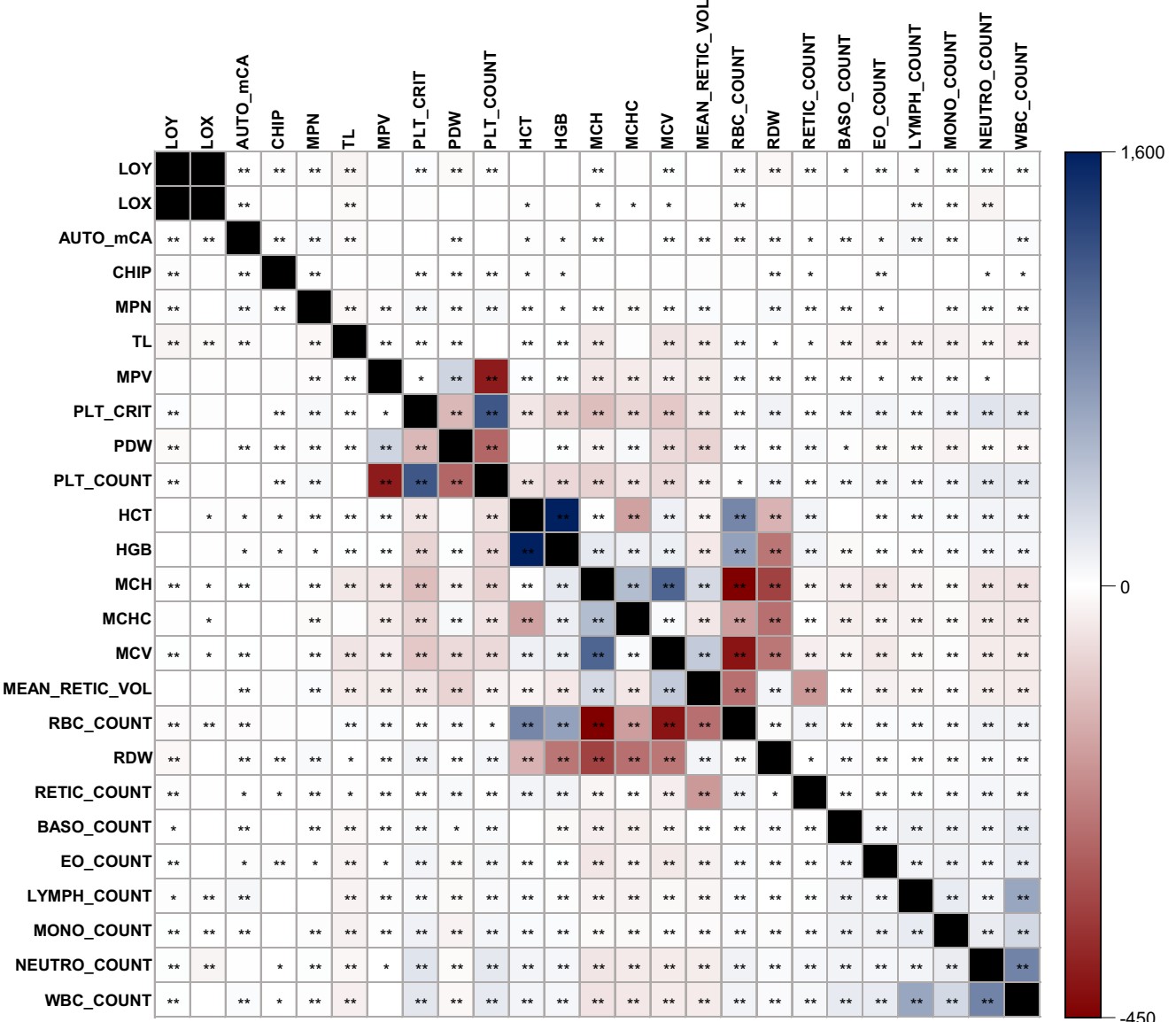

**Fig. 3 | Pairwise phenotypic associations between each type of clonal hematopoiesis, telomere length, and 19 blood cell traits.** Blue, positive T-statistic; red, negative T-statistic. T-statistics were derived using linear regression adjusted for age, age-squared, 25-level smoking status, and sex (in non LOY or LOX comparisons). Black cells were not tested. T-statistics that are significantly different from zero at a nominal $p$-value ($p < 0.05$) are marked with an asterisk and Bonferroni corrected $p$-value ($p < 1.67 \times 10^{-4}$) are marked with two asterisks. All pairwise T-statistics and p-values are given in Supplementary Data 4.

stronger associations with lymphoid malignancy risk (HR = 16.43, 95% CI = 7.80–34.60, $P = 1.77 \times 10^{-13}$) and myeloid malignancy risk (HR = 63.46, 95% CI = 26.01–154.84, $P = 7.56 \times 10^{-20}$) (Supplementary Fig. 11 and Supplementary Data 13).

We also assessed incident hematological malignancy risk with all combinations of LOY, LOX, CHIP, and autosomal mCAs (Table 1). Individuals with only LOX (HR = 1.46, 95% CI = 1.03–2.01, $P = 2.09 \times 10^{-2}$) or only autosomal mCAs (HR = 5.24, 95% CI = 4.47–6.14, $P = 2.89 \times 10^{-93}$) had significant risk of incident lymphoid malignancies. Individuals with LOY or CHIP were only associated with increased lymphoid malignancy risk when also harboring autosomal mCAs (Table 1). For myeloid malignancies, individuals with only CHIP mutations (HR = 5.44, 95% CI = 3.97–7.47, $P = 9.00 \times 10^{-26}$) or only autosomal mCAs (HR = 7.19, 95% CI = 5.32–9.70, $P = 6.14 \times 10^{-38}$) were associated with increased risk. Individuals carrying LOY or LOX mutations were only associated with increased myeloid malignancy risk when also harboring CHIP or autosomal mCAs. The co-occurrence and overlap of CHIP and autosomal mCAs motivates future studies that jointly assess

both CH traits to better understand CH interactions that could confer increased propensity for clonal expansion and elevated disease and mortality risk, particularly at specific loci or with specific mutations[9].

Pathway-based analyses using GWAS summary statistics (Online Methods) utilized 6290 curated gene sets and canonical pathways from Gene Set Enrichment Analysis (GSEA) and revealed significant associations between several biological pathways and types of CH (Supplementary Fig. 12) with all types of CH associated with gene sets related to apoptosis, IL-2 signaling, DNA methylation, promyelocytic leukemia gene product (PML) targets, and cancer-related gene sets (Supplementary Data 14–18). LOY, LOX, and MPN were significantly associated with hematopoietic progenitor cells, hematopoietic cell lineage and differentiation gene sets, and DNA damage response (Supplementary Data 14, 15, and 18). LOY, autosomal mCAs, CHIP, and MPN were associated with telomere extension by telomerase, with LOY and MPN also associated with telomere stress-induced senescence (Supplementary Data 14, 16–18). Additionally, the 12 genetic pathways significantly associated with autosomal mCAs were also associated

**Table 1 | Incident lymphoid and myeloid malignancy associations (HR) with 95% confidence intervals by CH status**

| Malignancy | CH status | $N_{Cancer}$ | $N_{No\_cancer}$ | HR (95% CI) | p-value[a] |
|---|---|---|---|---|---|
| Lymphoid | | | | | |
| | No LOY, LOX, CHIP, or Autosomal mCAs | 919 | 163,193 | REF | – |
| | LOY only | 169 | 15,458 | 1.03 (0.86–1.23) | $7.67 \times 10^{-1}$ |
| | LOX only | 41 | 4620 | 1.46 (1.06–2.01) | $2.09 \times 10^{-2}$ |
| | CHIP only | 51 | 5837 | 1.19 (0.90–1.58) | $2.32 \times 10^{-1}$ |
| | Autosomal mCA only | 186 | 5198 | 5.24 (4.47–6.14) | $2.89 \times 10^{-93}$ |
| | LOY and CHIP | 7 | 727 | 0.86 (0.41–1.81) | $6.83 \times 10^{-1}$ |
| | LOY and Autosomal mCA | 40 | 787 | 4.38 (3.16–6.06) | $5.40 \times 10^{-19}$ |
| | LOX and CHIP | 1 | 218 | 0.69 (0.10–4.89) | $7.08 \times 10^{-1}$ |
| | LOX and Autosomal mCA | 20 | 267 | 10.93 (6.98–17.13) | $1.55 \times 10^{-25}$ |
| | CHIP and Autosomal mCA | 26 | 313 | 10.06 (6.80–14.89) | $6.55 \times 10^{-31}$ |
| | LOY, CHIP, and Autosomal mCA | 3 | 71 | 3.36 (1.08–10.49) | $3.64 \times 10^{-2}$ |
| | LOX, CHIP, and Autosomal mCA | 6 | 20 | 41.10 (18.34–92.09) | $1.77 \times 10^{-19}$ |
| Myeloid | | | | | |
| | No LOY, LOX, CHIP, or Autosomal mCAs | 202 | 163,910 | REF | - |
| | LOY only | 29 | 15,598 | 0.98 (0.65–1.49) | $9.34 \times 10^{-1}$ |
| | LOX only | 10 | 4651 | 1.48 (0.78–2.83) | $2.30 \times 10^{-1}$ |
| | CHIP only | 49 | 5839 | 5.44 (3.97–7.47) | $9.00 \times 10^{-26}$ |
| | Autosomal mCA only | 55 | 5329 | 7.19 (5.32–9.7) | $6.14 \times 10^{-38}$ |
| | LOY and CHIP | 9 | 725 | 6.14 (3.09–12.2) | $2.19 \times 10^{-7}$ |
| | LOY and Autosomal mCA | 4 | 823 | 2.45 (0.9–6.67) | $7.96 \times 10^{-2}$ |
| | LOX and CHIP | 0 | 219 | - | $9.94 \times 10^{-1\,b}$ |
| | LOX and Autosomal mCA | 1 | 286 | 2.21 (0.31–15.86) | $4.29 \times 10^{-1}$ |
| | CHIP and Autosomal mCA | 17 | 322 | 31.69 (19.2–52.28) | $1.10 \times 10^{-41}$ |
| | LOY, CHIP, and Autosomal mCA | 1 | 73 | 6.42 (0.89–46.15) | $6.47 \times 10^{-2}$ |
| | LOX, CHIP, and Autosomal mCA | 3 | 23 | 87.29 (27.67–275.35) | $2.45 \times 10^{-14}$ |

Associations were derived using Cox proportional hazards regression adjusted for age, age-squared, 25-level smoking status, and sex.
*CH* clonal hematopoiesis, *HR* hazard ratio, *CI* confidence interval.
[a]Two-sided Cox proportional hazards regression.
[b]There were no incident myeloid cancers in this group.

with CHIP, providing further evidence that these types of CH are interrelated. Overall, pathway analyses suggest core shared pathogenic mechanisms related to cellular differentiation, DNA damage repair, and cell cycle regulation that are critical for the development and clonal expansion of most types of CH.

**Correlation of types of CH with myeloid and lymphoid cell traits**
We examined genetic and phenotypic correlations between types of CH and 19 blood cell traits to assess lineage-specific effects by type of CH (Figs. 2, 3). All types of CH displayed positive genetic correlations for both plateletcrit ($P < 0.02$) and platelet count ($P < 0.05$) (Fig. 2, Supplementary Data 1). LOY, MPN, and CHIP were the only types of CH to also display significant phenotypic associations with plateletcrit ($P < 2 \times 10^{-13}$) and platelet count ($P < 1.5 \times 10^{-13}$) (Fig. 3, Supplementary Data 4). LOY and MPN demonstrated additional genetic correlations enriched for myeloid lineage traits, namely, positive correlations with total white blood cell, eosinophil, monocyte, and neutrophil counts ($P < 0.026$; Fig. 2, Supplementary Data 1). LOY was additionally positively correlated with lymphocyte count ($\rho = 0.05$, $P = 8.74 \times 10^{-3}$; Fig. 2, Supplementary Data 1). MPN was positively correlated with other myeloid lineage traits including hematocrit, hemoglobin, and red blood cell count ($P < 6.5 \times 10^{-4}$; Fig. 2, Supplementary Data 1), as previously reported[22]. In support of the genetic correlations, we observed strong phenotypic associations of LOY and MPN with myeloid traits that closely mirror the magnitude and significance of the genetic correlation results (Fig. 3, Supplementary Data 4), and previously reported phenotypic associations[23–25]. Both LOY and MPN were

positively associated with monocyte, neutrophil, and white blood cell counts ($P < 1.5 \times 10^{-41}$; Fig. 3, Supplementary Data 4). LOY was also inversely associated with lymphocyte count (T-statistic = −2.75, $P = 5.91 \times 10^{-3}$) (Fig. 3, Supplementary Data 4). These findings suggest shared mechanisms regulating hematopoiesis likely also govern susceptibility to LOY and MPN.

LOX was the only type of CH to display both a positive genetic correlation ($\rho = 0.17$, $P = 8.40 \times 10^{-5}$; Fig. 2, Supplementary Data 1) and a positive phenotypic association with lymphocyte count (T-statistic = 23.96, $P = 9.11 \times 10^{-127}$) (Fig. 3, Supplementary Data 4). LOX had a positive genetic correlation with myeloid traits such as basophil count and eosinophil count, whereas it displayed an inverse genetic correlation with hematocrit and hemoglobin ($P < 0.02$, Fig. 2, Supplementary Data 1). LOX also had positive phenotypic associations with MCH, MCHC, MCV, and monocyte count ($P < 0.015$, Fig. 3, Supplementary Data 4), and inverse associations with hematocrit, red blood cell count, and neutrophil count ($P < 0.03$, Fig. 3, Supplementary Data 4).

Besides the aforementioned genetic correlations with plateletcrit and platelet count, we observed additional genetic correlations between autosomal mCAs and CHIP with blood cell traits (Fig. 2). Inverse genetic correlations were observed between autosomal mCAs with MCH, MCV, and mean reticulocyte volume ($P < 3.0 \times 10^{-2}$, Fig. 2, Supplementary Data 1). CHIP had a positive genetic correlation with MCHC and reticulocyte count ($P < 4.0 \times 10^{-2}$, Fig. 2, Supplementary Data 1). In the case of combined autosomal mCAs, there was evidence for positive phenotypic associations with both lymphocyte count (T-statistic = 60.33, $P < 5 \times 10^{-324}$) and total white blood cell count (T-

statistic= 34.48, $P = 3.89 \times 10^{-260}$) (Fig. 3, Supplementary Data 4). Combined CHIP was positively associated with platelet distribution width (T-statistic = 5.02, $P = 5.26 \times 10^{-7}$), red blood cell distribution width (T-statistic = 4.09, $P = 4.37 \times 10^{-5}$), and neutrophil count (T-statistic = 3.59, $P = 3.32 \times 10^{-4}$) (Fig. 3, Supplementary Data 4), all of which are myeloid lineage traits. The CHIP phenotypic association findings support recent evidence suggesting CHIP primarily results in myeloid-related disruptions, although select distinct CHIP events could increase the risk for disruptions in the lymphoid lineage[9]. Together our results support lineage-specific effects that differ by type of CH, suggesting shared etiology, specifically shared genetic etiology for LOY and myeloid traits, as well as ample phenotypic associations that detail early downstream phenotypic disruptions in hematologic phenotypes that alter disease risk.

## A dynamic association of telomere length with CH

Telomere length in leukocytes provides a metric of hematopoietic stem cell activity and can provide insights into how genetic variation in hematopoietic stem cells interact with risk for acquiring CH[22,26]. The genetic relationship between each type of CH with leukocyte telomere length (TL) was evaluated to determine whether genetic variation in telomere maintenance genes could also contribute to predisposition to CH. A positive genetic correlation for autosomal mCAs with TL was observed ($\rho = 0.23$, $P = 4.95 \times 10^{-3}$) (Fig. 2, Supplementary Data 1). To further test for a causal relationship with TL, we conducted one-direction Mendelian randomization (MR) between TL and each CH type using 130 previously published TL-associated variants (Supplementary Fig. 13)[27]. Based on MR-IVW models, we observed a positive relationship between the TL IV and autosomal mCAs ($Z_{filtered} = 5.65$, $P = 1.21 \times 10^{-7}$), CHIP ($Z_{filtered} = 5.72$, $P = 9.65 \times 10^{-8}$), and MPNs ($Z_{filtered} = 5.61$, $P = 1.88 \times 10^{-7}$), and observed a negative relationship between the TL IV and LOY ($Z_{filtered} = -6.40$, $P = 8.11 \times 10^{-9}$) and did not identify evidence for a causal relationship between telomere length and LOX (Supplementary Fig. 14, Supplementary Data 19). These observations provide additional support of a causal relationship between inherited telomere length and select CH traits[12,22,28–31]. The intercept from MR-Egger regression was significant ($p < 0.05$) for both autosomal mCAs and MPN (Supplementary Data 19), so we performed additional MR weighted median (MR-WM) analyses which displayed the same positive relationships between the TL IV and autosomal mCAs ($Z_{filtered} = 4.16$, $P = 6.19 \times 10^{-5}$), and MPNs ($Z_{filtered} = 4.34$, $P = 3.53 \times 10^{-5}$) (Supplementary Data 19). These MR relationships are supported by our pathway analyses, which demonstrate telomere pathways are significantly associated with LOY, autosomal mCAs, CHIP, and MPN (Supplementary Data 14, 16–18). Based on these data it is plausible that inherited variation in telomere length maintenance contributes to clonal expansion of mutated hematopoietic stem cells, or alternatively confers greater risk for mutation acquisition and clonal evolution in hematopoietic stem cells.

Measured telomere length is a metric of hematopoietic stem cell growth and clonal expansion. Using available measured telomere length data from UK Biobank, we observed inverse phenotypic associations between CH and measured telomere length (Fig. 3). CHIP, which presents with the smallest fraction of mutated clones, had an insignificant phenotypic association with measured TL (T-statistic = $-1.03$, $P = 0.30$) (Fig. 3, Supplementary Data 4). To further examine this relationship, we conducted analyses between CHIP VAF and measured TL and observed individuals with higher VAF, i.e., higher CHIP cellular fraction, had a more inverse association with measured TL (T-statistic = $-6.50$, $P = 8.34 \times 10^{-11}$, Supplementary Fig. 15a and Supplementary Data 20). Additionally, individuals with higher autosomal mCA cellular fraction also demonstrated a stronger inverse association with measured TL (T-statistic = $-9.02$, $P = 2.16 \times 10^{-19}$, Supplementary Fig. 15b and Supplementary Data 21). The number of mutations present in an individual was also inversely associated with TL for increasing autosomal mCA count (T-statistic = $-10.01$, $P = 1.35 \times 10^{-23}$). Individuals with both CHIP and autosomal mCAs demonstrated an inverse association with TL (T-statistic = $-2.75$, $P = 5.93 \times 10^{-3}$) as well, with individuals carrying overlapping CHIP and autosomal mCAs displaying a stronger inverse association with TL (T-statistic = $-3.48$, $P = 5.01 \times 10^{-4}$) compared to individuals without CHIP or autosomal mCAs (Supplementary Fig. 16 and Supplementary Data 22). These inverse TL associations indicate increased clonal expansion leads to reduced measured telomere length and suggest reductions in telomere length from the expansion of mutated clones could lead to further genomic instability and the acquisition of additional CH mutations.

## Leveraging shared correlations to nominate additional MPN susceptibility loci

Finally, we leveraged the shared genetic architecture between these CH traits (Fig. 4) to identify novel loci associated with MPN - a disease where it has been challenging for GWAS to perform well powered case-control analyses, despite the finding of considerable heritable influences on this disorder[22]. We first performed multi-trait analysis of GWAS (MTAG), which boosts the power to identify potential MPN-associated signals by leveraging the shared genetic architecture with LOY and TL (Online Methods). This approach identified 25 MPN loci at genome-wide significance ($P < 5 \times 10^{-8}$), 15 of which have not been previously implicated in MPN (Supplementary Data 23). We next evaluated a complementary approach of performing colocalization analyses (Online Methods) using genome-wide significant loci associated with LOY, TL, and MPN. We found that 12 LOY loci, mapping to 11 genes (TET2, NREP, GFI1B, TERT, DLK1, PARP1, TP53, RBPMS, MAD1L1, MECOM, and ATM) co-localized with MPN (Supplementary Data 24), highlighting 6 loci that have not previously reached genome-wide significance for MPN ($P = 1.17 \times 10^{-4}$ to $5.14 \times 10^{-8}$). In addition, 5 leading SNPs for TL co-localized with MPN and mapped to 4 genes (TERT, NFE2, PARP1, and ATM), 2 of which have not reached genome-wide significance ($P < 5 \times 10^{-8}$) in prior MPN analyses (NFE2 and PARP1) (Supplementary Data 25). Of note, leading SNPs at TERT, PARP1, and ATM colocalized across all 3 traits (Supplementary Data 24 and Supplementary Data 25), and 5 co-localized loci also reached genome-wide significance in the MTAG analysis (PARP1, MAD1L1, DLK1, RBPMS, and TP53) (Fig. 5). While validation is required for the newly identified putative MPN risk loci, these results illuminate opportunities to use insights from correlated diseases or phenotypes to gain new genetic and biological insights on blood cancer risk.

## Discussion

Understanding the underlying molecular mechanisms of different types of CH is critical for disentangling age-related clonal evolution and the possible impact of CH on future disease risk, particularly the risk of acquiring hematologic malignancies. We performed one of the first systematic analyses to examine associations across all types of CH using large-scale genetic and phenotypic data. Our results highlight both similarities in the underlying mechanisms and key differences, particularly with respect to distinct aspects of hematopoiesis. Common to the types of CH are core pathways, namely, cellular differentiation, DNA repair, and cell cycle regulation, that contribute to the generation and clonal expansion of CH types. Together, these findings detail specific characteristics of CH that should be investigated to improve the utility of detectable CH for disease risk and possible intervention or prevention.

Prior reports have focused on phenotypic relationships between types of CH, not genetic associations, and could be missing important relationships relevant for informing mechanism. We provide evidence for genome-wide genetic correlations between LOY and LOX, LOY and MPN, autosomal mCAs and MPN, as well as between CHIP and MPN, suggesting shared biologic mechanisms promoting or predisposing to

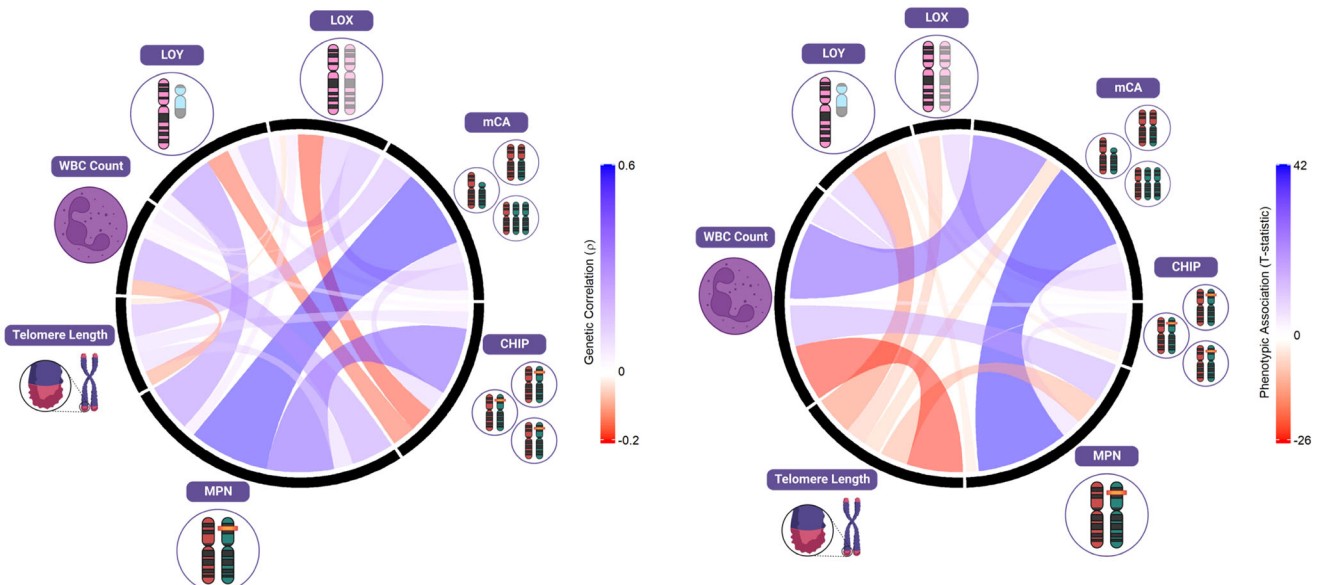

**Fig. 4 | Shared etiologies and associations between types of clonal hematopoiesis (CH) and hematopoietic phenotypes: telomere length, and white blood cell count (WBC).** Pairwise high-definition likelihood (HDL) genetic correlations are given in the left plot, pairwise phenotypic associations derived using linear regression adjusted for age, age-squared, 25-level smoking status, and sex (in non-LOY or LOX comparisons) are given in the right plot. The black lines separate types of CH from each other and the color and width of the bands represent the strength of association. All genetic correlations are available in Supplementary Data 1 and 4.

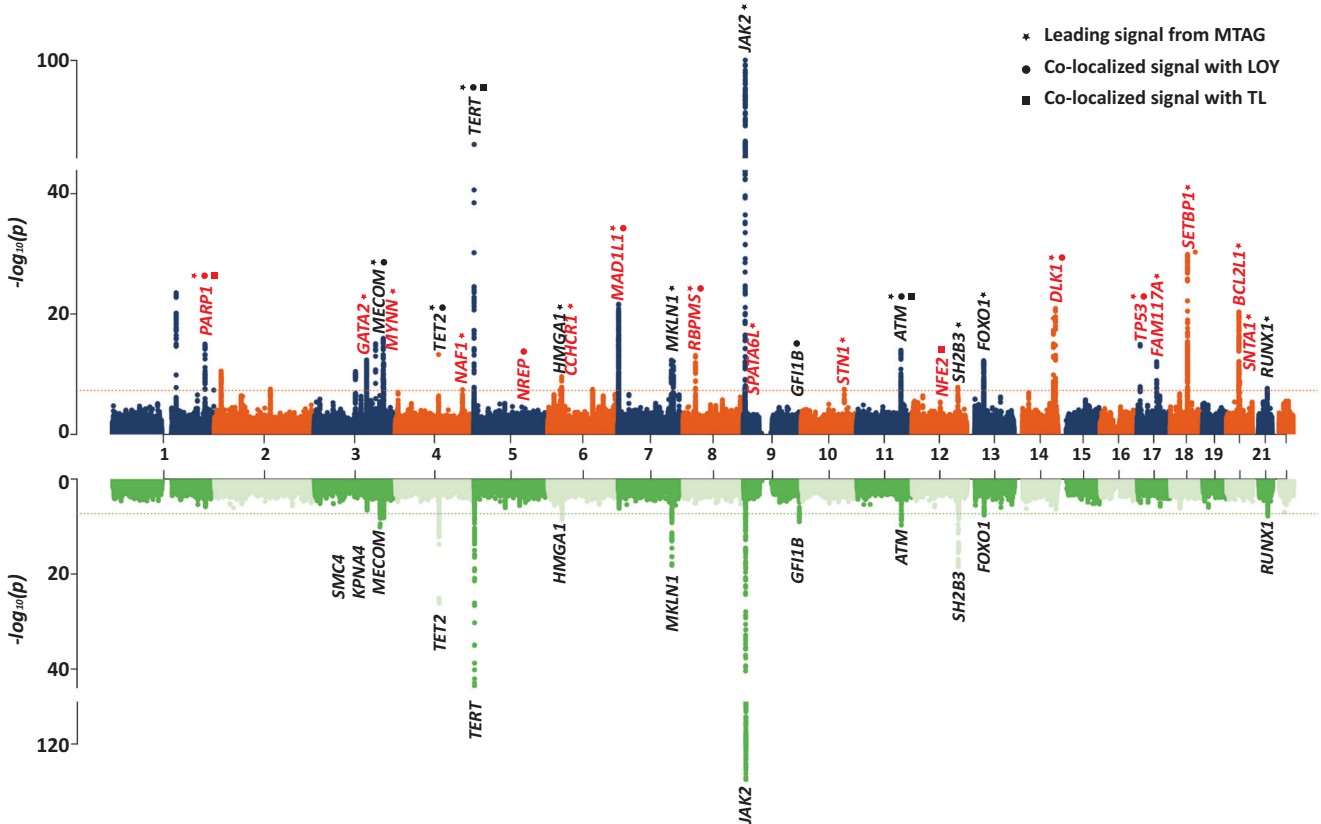

**Fig. 5 | Stacked Manhattan plots from the Multi-Trait Analysis of genome-wide association study (GWAS) summary statistics (MTAG) and colocalization analyses among telomere length, LOY, and MPN (Top plot) and the original MPN GWAS performed by Bao et al. (Bottom plot).** Nominated MPN susceptibility loci are labeled by analysis, MTAG(stars, ★), loss of chromosome Y (LOY) colocalization (circles, ●), TL colocalization (squares, ■), and colored red for previously unidentified and black for previous identified. Supporting data for MTAG signals are defined in Supplementary Data 23 and data for colocalization analyses is available in Supplementary Data 24 and 25.

the development and clonal expansion of different CH types. Likewise, these genetic relationships can be leveraged to identify disease susceptibility loci for related traits, as we demonstrated for MPN. Genetic correlations with blood cell traits further demonstrate lineage-specific effects that differ by type of CH. As many of these CH and blood cell traits are interrelated, we report associations that do not adjust for stringent multiple testing corrections and caution against the over-interpretation of marginally significant associations.

Phenotypic associations between types of CH provide additional evidence for a shared genetic etiology, but could also indicate shared environmental factors that drive CH growth and expansion (e.g., medication[32,33] or tobacco usage[12,34–38]). A notable discordance in directionality is between the genetic correlation and the phenotypic relationship between LOY and MPN; although further genetic investigations in larger MPN sets are needed to replicate these findings. This potential discordance supports a shared genetic etiology as indicated by the genetic associations, but the phenotypic relationship indicates a mutual exclusivity between LOY and MPN suggesting that for some types of CH when one type of CH develops, the occurrence of others could be suppressed (e.g., *DNMT3A* and *TET2* CHIP[39]). This observed mutual exclusivity is most likely due to hematopoietic stem cells being unable to tolerate multiple independent somatic drivers of CH. Individuals with CHIP and higher cellular fraction autosomal mCAs also demonstrated an inverse phenotypic association with LOY, indicating a similar mutually exclusive relationship. Our findings support common genetic factors across types of CH and raise the importance of pursuing shared environmental contributors beyond smoking and air-pollution that could differentially impact clonal selection of CH types[12,34–38,40].

While our findings reveal possible mutual exclusivity of CH, we also observe strong evidence for the co-occurrence of CHIP and autosomal mCAs in the same individual, and in many instances, overlapping within known CHIP driver mutations (e.g., *TET2, DNMT3A, JAK2*). Cross-sectional observations of cellular fraction indicate the CHIP mutations often precede autosomal mCAs, which can lead to preferential clonal expansion of mCAs containing CHIP mutations (i.e., a "second hit"), as has been mechanistically examined in specific cases[41].

Rapid clonal expansion afforded by each type of CH leads to marked reductions in measured telomere length. These reductions in telomere length can lead to increased genomic instability in individuals with CH and could increase the likelihood of acquiring additional types of CH. Individuals who acquired overlapping CHIP and autosomal mCAs were found to have greater reductions in telomere length, which is a marker of past clonal expansion, while also having a significantly higher risk for acquiring hematologic malignancies. Detection of these highly clonal, co-occurring CH events, especially at *TET2, DNMT3A,* and *JAK2*, could be helpful in identifying individuals at increased risk of developing hematologic malignancies. Future studies should focus on investigating the co-occurrence and overlap of CHIP and autosomal mCAs to further evaluate associations with environmental factors, elevated disease risk, and mortality.

## Methods

### Ethics statement
Research conducted herin complies with all relevant ethical regulations. UK Biobank has approval from the Northwest Multi-centre Research Ethics Committee (MREC) as a Research Tissue Bank (RTB) approval. All participants provided informed consent to paritcipate in the UK Biobank.

### Hematopoietic phenotypes
We used genome-wide association study (GWAS) summary statistics to investigate germline similarities and differences of 25 hematopoietic-related phenotypes. These included loss of chromosome Y (LOY) in

men[13], loss of chromosome X (LOX) in women, autosomal mCAs including gains, losses, and copy neutral loss of heterozygosity[30], CHIP, MPNs[22], leukocyte telomere length (TL)[27], and 19 blood cell traits[22] (Supplementary Data 26). For LOX, we used previously generated data on copy number variation[25,30], and performed GWAS on 243,106 women in the UK Biobank using a linear mixed model implemented in BOLT-LMM[42], to account for cryptic population structure and relatedness, a similar methodology was used to conduct the LOY GWAS[13]. For CHIP, we called somatic mutations using Mutect2 from available UK Biobank 200K whole exome sequencing data[43]. A QCed set of $N = 198,178$ individuals were analyzed using a panel of one hundred normals created from UK Biobank participants with age <40, and included as part of the QC process. Variants were considered passing QC if the following criteria were met: meeting FilterMutectCalls quality standards including learned read orientation, variant allele fraction (VAF) >=2%, depth of calling >10, and a Phred scaled GERMQ score of >20 (1% error rate). CHIP was defined in these individuals using a curated list of CHIP mutated variants as previously described in the UKBB WES cohort (Supplementary Data 27)[44]. Individuals with a diagnosis of myeloid malignancy (AML, MDS, MPN) before blood draw were excluded from the CHIP phenotype while those that went on to develop myeloid malignancy post-blood draw by at least 5 years were retained. In total, 7280 (3.7%) individuals were found to have at least one CHIP curated variant (Supplementary Data 27, Supplementary Data 28). We performed a CHIP GWAS in the UK Biobank array data[45], restricted to European ancestry individuals (Supplementary Data 26) and those passing the following QC measures: individual had not withdrawn consent, included in kinship inference, no excess (>10) of putative third-degree relatives inferred from kinship, not an outlier in heterozygosity and missing rates, not found to have putative sex chromosome aneuploidy, no genotype missing rate of >0.1. Variants were included if they had a genotype missing rate <0.1 across QC'ed individuals, Hardy-Weinberg equilibrium $p$-value of $>1 \times 10^{-15}$, and a minor allele frequency of >0.1%. Using Regenie[45], we performed ridge regression in step 1 using a set of ~300,000 pruned SNPs and default cross-validation settings. We included age, age-squared, sex, smoking status (categorical level variable), and principal components of ancestry 1 through 10 as covariates. To further increase the power of downstream analyses, we conducted a meta-analysis using METAL[46] of CHIP GWAS summary statistics between our generated UK Biobank summary statistics and those from a previous CHIP GWAS in TOPMed[12]. Before meta-analysis, both GWAS summary statistics were lifted-over from their respective genome builds to reference genome hg19. Alleles were flipped according to the hg19 build reference allele, and if neither allele was present the variant was removed. Strand ambiguous and non-biallelic SNPs were removed. Minor allele frequency was filtered to > = 0.1%. RSIDs were assigned to variants using dbSNP version 144.

Data on 482,378 subjects from the UK Biobank were used for phenotypic association analyses, after removing individuals with sex discordance or whose DNA failed genotyping QC. We used previously generated data for each of the hematopoietic related phenotypes[13,22,27,30], with the exception of CHIP which was called in the UK Biobank as detailed above (Supplementary Data 3).

### Genetic correlation
We used both the high-definition likelihood method[47] (HDL) and linkage disequilibrium score regression[48] (LDSC) to compute pairwise genetic correlation between hematopoietic phenotypes. For HDL, we utilized an LD score reference panel available within HDL which contains 1,029,876 QCed UK Biobank imputed HapMap3 SNPs[47], and also calculated the observed heritability for each hematopoietic phenotype (Supplementary Fig. 17 and Supplementary Data 29). For LDSC, we utilized an LD score reference panel generated on 6285 European ancestry individuals combined from the 1000 Genomes Phase 3 and

UK10K cohorts, with a total of 17,478,437 available variants, and GWAS summary statistics were filtered to include overlapping variants with >1% MAF and >90% imputation quality score. As all of the included GWAS summary statistics utilized UK Biobank data, we constrained the intercept within both HDL and LDSC by accounting for both the known sample overlap and phenotypic correlation between traits (Supplementary Data 30). Pairwise genetic correlations with CHIP were conducted using the generated UK Biobank summary statistics, due to incomplete overlap with the HDL reference panel within the TOPMed CHIP summary statistics (Supplementary Data 26 and Supplementary Data 30). Correlation matrixes and circular charts for results visualization were generated using the "corrplot" (v0.92) and "circlize" (0.4.15) R packages[49,50].

## Phenotypic associations

Pairwise phenotypic associations between all hematopoietic phenotypes were generated using linear regression adjusting for age, age-squared, sex (in non sex-specific traits), and a 25-level smoking variable to reduce the potential for confounding variables driving associations[36]. To ensure compatibility between binary and continuous phenotypic association results, association T-statistics were generated and reported to measure strength and direction of phenotypic associations.

## Hematological malignancy association

Using available data within UK Biobank, we extracted relevant cancer information from both inpatient records and cancer registry data. Incident hematological cancers were defined as occurring after study enrollment. Lymphoid cancers were defined using codes: C81: Hodgkin's disease, C82: Follicular [nodular] non-Hodgkin's lymphoma, C83: Diffuse non-Hodgkin's lymphoma, C84: Peripheral and cutaneous T-cell lymphomas, C85: Other and unspecified types of non-Hodgkin's lymphoma, C86: Other specified types of T/NK-cell lymphoma, C88: Malignant immunoproliferative diseases, C90: Multiple myeloma and malignant plasma cell neoplasms, C91: Lymphoid leukemia, C96.3: True histiocytic lymphoma, D47.2 Monoclonal gammopathy. Myeloid cancers were defined using codes: C92: Myeloid leukemia, C93: Monocytic leukemia, C94.0: Acute erythraemia and erythroleukaemia, C94.2: Acute megakaryoblastic leukemia, C94.4: Acute panmyelosis, C94.5: Acute myelofibrosis, C94.6: Myelodysplastic and myeloproliferative disease, not elsewhere classified, C96.0: Letterer-Siwe disease, C96.1: Malignant histiocytosis, C96.2: Malignant mast cell tumor, C96.5: Multifocal and unisystemic Langerhans-cell histiocytosis, C96.6: Unifocal Langerhans-cell histiocytosis, C96.8: Histiocytic sarcoma, D47.0: Histiocytic and mast cell tumors of uncertain and unknown behavior, D47.1: Chronic myeloproliferative disease, D47.3: Essential (hemorrhagic) thrombocythaemia, D47.4: Osteomyelofibrosis, D47.5: Chronic eosinophilic leukemia [hypereosinophilic syndrome].We performed Cox proportional hazards regression to assess the risk of hematological malignancies across CH groups adjusting for age, age-squared, sex, and a 25-level smoking variable[36]. Hazards ratios and 95% confidence intervals were generated and reported to measure strength and direction of hematological malignancy risk.

## Mendelian randomization analyses

We performed one-direction Mendelian randomization (MR) between TL and LOY, LOX, autosomal mCAs, CHIP, and MPN. Briefly, MR analyses utilize genetic variants from GWAS as instrumental variables (IVs) to assess the directional association between an exposure and outcome, which can mimic the biological link between exposure and outcome[51]. Each variant used in a MR analysis must satisfy three assumptions: (1) it is associated with the risk factor, (2) it is not associated with any confounder of the risk factor–outcome association, (3) it is conditionally independent of the outcome given the risk factor and confounders[52,53].

For our analyses, we used summary statistics and 130 significant signals of the largest TL GWAS to date to form the TL IV[27]. We then extracted the same set of signals from summary statistics for each CH outcome. If any signals were missing in the outcome summary statistics, we collected proxies for these signals using GCTA[54] with European UK Biobank individuals as reference (within 1 MB of reported signals and $R^2 > 0.4$). We chose the proxy of each missing signal with the largest $R^2$ value as the replacement IV, which was contained in both GWAS summary statistics of the exposure and outcome. All TL signals were aligned to increasing allele and alleles for outcome were realigned accordingly.

The MR inverse-variance weighted (MR-IVW) model, which can provide high statistical power[55], was used as our primary analysis. As some signals may have a stronger association with the outcome than the exposure, which may induce reverse causality, we applied Steiger filtering to each IV in order to remove these variants using the "TwoSampleMR" R package (v0.5.6)[56]. We then applied Radial filtering to exclude signals that were identified as outliers according to Rücker's Q' statistic[57].

The sensitivity of MR models was checked by the degree of heterogeneity ($I^2$ statistics and Cochran's Q-derived P-value), horizontal pleiotropy (MR-Egger $p_{intercept} < 0.05$), and funnel and dosage plots (Supplementary Fig. 13). To account for potential horizontal pleiotropy and heterogeneity, three additional MR models were performed: MR-Egger[58], weighted median (MR-WM)[59], and penalized weighted median (MR-PWM)[59].

## Pathway and Gene Set Analyses

We performed agnostic pathway-based analyses using the summary data-based adaptive rank truncated product (sARTP) method, which combines GWAS summary statistics across SNPs in a gene or a pathway[60], to identify gene sets and pathways associated with each type of CH. A total of 6,290 curated gene sets and canonical pathways from GSEA (https://www.gsea-msigdb.org/gsea/msigdb/) were used for the analyses. For each type of CH, the signals from up to five of the most associated SNPs in a gene were accumulated. We adjusted for the number of SNPs in a gene and the number of genes in a pathway through a resampling procedure that controls for false positives[60]. The P values of gene- and pathway-level associations were estimated from the resampled null distribution generated from 100 million permutations. Linkage disequilibrium between SNPs was computed from European individuals within 1000 Genomes Project data[61]. To reduce the potential for population stratification to bias the results, we rescaled the marginal SNP results for each CH trait to set the genomic control inflation factor to 1. A Bonferroni corrected level of significance of $7.95 \times 10^{-6}$ (0.05/6,290 GSEA pathways) was used to assess statistical significance.

## Rare variants gene-burden test for LOY in UK Biobank

To explore the relationship between rare variant burden and LOY, we performed association tests using whole exome sequencing (WES) data for 190,759 males provided by the UK Biobank. Prior to performing association tests, we performed quality control on provided sequencing data as previously described[62].

We utilized the ENSEMBL Variant Effect Predictor (VEP) v104[63] to annotate variants on the autosomal and X chromosomes. VEP was run with default settings, the "everything" flag, and the LOFTEE plugin[64]. The predicted consequence of each variant was prioritized by a single MANE (version:0.97) or, when not available, a VEP canonical ENSEMBL transcript, and the most damaging consequence as defined by VEP defaults. Variants with high confidence (HC, as defined by LOFTEE) stop gained, splice donor/acceptor, and frameshift consequences were grouped as protein-truncating variants (PTVs). Following transcript annotation, we utilized CADD v1.6 to calculate the Combined Annotation Dependent Depletion (CADD) score for each variant[65].

To perform gene burden tests, we implemented BOLT-LMM v2.3.6[42]. As input, BOLT-LMM requires genotyping data for variants with allele count greater than 100, all variants from WES passing QC as defined above, and a set of dummy genotypes representing participant carrier status per-gene for PTVs, missense variants with CADD ≥ 25 (MISS_CADD25,) and damaging variants (HC_PTV + MISS_CADD25, DMG). Dummy genotypes were generated by collapsing all variants within each gene with a minor allele frequency (MAF) < 0.1%. For each gene, carriers with a qualifying variant were set to heterozygous ("0/1") and non-carriers were set as homozygous reference ("0/0"). All models were controlled for age, age-squared, WES batch, and the first ten genetic ancestry principal components (PCs) as generated by Bycroft et al.[66].

Following association testing, we further excluded genes with less than 50 non-synonymous variant carriers, leaving 8984 genes of PTVs, 14,685 genes of MISS_CADD25, and 16,066 genes of DMG for an exome-wide significance threshold of $1.26 \times 10^{-6}$ (0.05/39,735) after Bonferroni correction (Supplementary Data 6).

### Associations between CHIP loss of function variant carriers and LOY

As associations between known CHIP genes and LOY identified as part of rare variant burden testing could be due to reverse-causality – somatic instability such as LOY could lead to, or occur in parallel with, variants arising within CHIP genes – we queried underlying variant call data to determine if individual variants within these genes were likely to have arisen somatically. We first extracted the number of reads supporting the alternate and reference alleles for all carriers of protein truncating variants (PTV) at MAF < 0.1% in four genes associated with LOY–three known CHIP genes, *ASXL1* ($n = 213$ carriers), *DNMT3A* ($n = 89$), and *TET2* ($n = 193$), and one control gene not previously associated with CHIP, *GIGYF1* ($n = 81$; Supplementary Fig. 3). This information was then used to calculate a Variant Allele Fraction (VAF) for each genotype, where a VAF of 0.5 indicates perfect balance between sequencing reads supporting the reference and alternate allele (Supplementary Fig. 4). For all variants, we also annotated whether it was found in a list of known, specific CHIP driver mutations or was likely to be a CHIP driver mutation based on a broader set of criteria presented in Bick et al.[12]. For each gene, we tested for an association between PTV carrier status and PAR-LOY except using 6 additional criteria that excluded individuals carrying:

1. Frameshift InDels with a binomial test *P*-value for allele balance <0.001 (i.e. filtering InDels identically to SNVs, see above).
2. Any variant with VAF < 0.25 or > 0.75.
3. Any variant with VAF < 0.4 or > 0.6.
4. Any variant with VAF > 0.35.
5. A variant explicitly listed in Supplementary Data 3 from Bick et al.[12]
6. A variant explicitly listed in Supplementary Data 3 or matching the criteria in Supplementary Data 2 from Bick et al.[12]

All association tests were run separately for each gene using a logistic model corrected for identical covariates as the rare variant burden tests outlined above.

### MTAG and colocalization analysis among TL, LOY, and MPN

GWAS summary statistics for LOY[13], TL[27], and MPN[22] were utilized to conduct a meta-analysis by implementing the multi-trait analysis of GWAS (MTAG)[67,68]. Based on the summary statistics from GWAS of multiple correlated traits, MTAG can enhance the statistical power to identify genetic associations for each trait included in the analysis[67,68]. We performed the MTAG analysis using Python (2.7.18). Prior to the analysis, we excluded the variants with MAF < 0.01 from the summary statistics of all three traits[13,22,27]. A potential problem for MTAG is that SNPs can be null for one trait but non-null for another trait, which can cause MTAG's effect size estimations of these SNPs for the first trait to shift away from 0. This causes the false positive rate (FDR) to increase. Therefore, we estimated the max FDR for each trait by invoking "−fdr" when running MTAG. We implemented a clumping algorithm to select signals from the MTAG generated MPN summary statistics. Preliminary leading signals were selected with a $P < 5 \times 10^{-8}$ and a MAF > 0.1% at a 1 Mb window. We then selected the secondary leading signals using approximate conditional analyses in GCTA[69] with UK Biobank reference panel. If the genome-wide significant leading signals shared any correlation with each other due to the long-range linkage disequilibrium ($r^2 > 0.05$), these signals were excluded from further analysis. We mapped the leading signals to the genes with 1 Mb window based on the start and end sites of genes' GRCh37 coordinates. For all leading signals, we extracted their summary statistics from the original MPN GWAS summary statistics. In total, 36 independent leading signals were identified. We then applied Bonferroni correction for the identified signals. We further excluded the signals with $P > 0.05/36 = 1.39 \times 10^{-3}$ in the original GWAS to avoid false positives mentioned above, as GWAS for both LOY and TL identified many more leading signals than MPN, which increased the FDR for MPN (Max FDR of MPN = 0.11).

We conducted the Bayesian test for colocalization between pairs of TL and MPN, and LOY and MPN using their summary statistics[13,22,27] and the leading GWAS signals by implementing R (3.6.3) package coloc (v5.1.0)[70]. The signals with posterior probability (h4.pp) ≥ 0.75 were defined as the co-localized causal variant for both traits. Manhattan plots for results visualization were generated using the "qqman" (0.1.4) R package[71].

### Reporting summary
Further information on research design is available in the Nature Portfolio Reporting Summary linked to this article.

### Data availability
All data used in the analysis is available from UK Biobank upon request (https://www.ukbiobank.ac.uk). Source data for figures are available in the Supplementary Data.

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

## Acknowledgements

This work was conducted using the UK Biobank resource (applications 9905, 21552, and 31063). The UK Biobank was established by the Wellcome Trust, the Medical Research Council, the United Kingdom Department of Health, and the Scottish Government. The UK Biobank has also received funding from the Welsh Assembly Government, the British Heart Foundation, and Diabetes UK. The clonal hematopoiesis infographic and models were created with BioRender.com. We thank Dr. Jacqueline B. Vo for her graphical support. The opinions expressed by the authors are their own and this material should not be interpreted as representing the official viewpoint of the U.S. Department of Health and Human Services, the National Institutes of Health or the National Cancer Institute. This work was supported by the intramural research program of the Division of Cancer Epidemiology and Genetics, National Cancer Institute, National Institutes of Health (D.W.B, A.K.H., A.D., T.R., L.S., K.Y., S.J.C., M.J.M.). J.R.B.P, Y.Z. and E.J.G. are supported by the Medical Research Council (Unit programs: MC_UU_12015/2, MC_UU_00006/2). S.K.N. is a Scholar of the American Society of Hematology. This work was also supported by the New York Stem Cell Foundation (V.G.S.), a gift from the Lodish Family to Boston Children's Hospital (V.G.S.), and National Institutes of Health Grants R01 DK103794, R01 CA265726, R01 HL146500 (V.G.S.). V.G.S. is a New York Stem Cell-Robertson Investigator.

## Author contributions

J.R.B.P, V.G.S. and M.J.M. conceived the study. L.D.C. and V.G.S. carried out the CHIP calls. D.W.B., L.D.C., Y.Z., E.L.B., E.J.G., A.K.H., A.D., T.R., L.S., K.Y. performed computational and statistical analyses. J.R.B.P., V.G.S. and M.J.M. supervised the study. D.W.B., L.D.C., Y.Z., S.K.N., S.J.C., J.R.B.P., V.G.S., A.K.H. and M.J.M. drafted the manuscript with input from all authors. All authors critically read and approved the final version of the manuscript.

## Funding

## Competing interests

V.G.S. serves as an advisor to and/or has equity in Branch Biosciences, Ensoma, Novartis, Forma, and Cellarity, all unrelated to the present work. All other authors declare no relevant competing interests.
