## [Peer Review File · Nature Communications]

Shared and distinct genetic etiologies for different types of clonal hematopoiesisEditorial Note: This manuscript has been previously reviewed at another journal that is not operating a transparent peer review scheme. This document only contains reviewer comments and rebuttal letters for versions considered at *Nature Communications*.

REVIEWER COMMENTS

Reviewer #1 (Remarks to the Author):

The authors have provided an incredibly thorough response effectively addressing all my previous questions and concerns as Reviewer #1, and have added additional analyses that add extra value to a paper that was already a substantial contribution to the field. I have no further comments.

Siddhartha Kar

Reviewer #2 (Remarks to the Author):

The manuscript from Brown et al. is improved. The CHIP variants identified are now more in line with prior studies, which adds confidence to the results. The use of HDL also identified novel genetic associations, although most of these are expected. I remain unconvinced of the LOY association results. For example, the inverse signal in Table S5 seems to be restricted largely to DNMT3A and TET2 mutations, and is absent in other genes, which has no apparent reason or explanation. A second issue is that the sensitivity to detect CHIP vs mCA is very different. It is not explicitly stated here, but the MOCHA calls in UKB have a median cell fraction of ~2-4% (equivalent to VAF of 1-2%). The median VAF in the CHIP calls would probably be ~8-9%. Thus, the CHIP clones are much larger than the mCA clones, which confounds the analyses. The authors should repeat association tests for mCA to CHIP using equivalent sensitivity thresholds, i.e. restricting mCA cell fraction to ~>15%. This would impact the interpretation. For example, in the current analysis they could conclude that having a large CHIP clone blocks the growth of other clones, including mCA, due to clonal competition. However, it does not allow for any inference on mutational cooperativity since most likely the events are in different clones. Restricting to only large mCA clones would be more likely to capture events that are co-occurring in the same cells to allow for better inference on cooperativity.

Minor:

The supplemental tables need better annotations and consistency. E.g. why is on the T-statistic reported by Table S5, not odds ratios or confidence intervals? I have no clue what Table S8-9 are meant to be. What do the column names mean in Table S10-11?

Reviewer #3 (Remarks to the Author):

In the revised paper by Brown et al., the authors elegantly refined the analyses, resulting in several additional findings. The HDL analysis revealed more correlations and generate further insights. Additional deep investigations supported the interesting inverse association between CHIP and LOY, as well as the CHIP-preceding-mCA hypothesis. The new analysis provided a comprehensive view of the combination of multiple CH types for the risk of hematological malignancies. The GWAS of MPN, empowered by shared genetic correlation of CH traits, identified 25 MPN loci, more than double the previous number.

The authors decided not to correct for multiple testing, but now included a caution message in the discussion, and discarded the term "significant" from some parts of the text, which is now appropriate I think.

The revised paper has resolved some conflicting results. However, the LOY-MPN association still shows opposite directions in genetic and phenotypic correlations. This reviewer believes that this may be due to the very low h^2 value of MPN sumstats, which led to an unreliable GC analysis result in this study. To support the results, if the authors intend not to remove genetic correlation studies about MPN, it might be better to perform analysis only with the effects of known significant

variants for comparison, in addition to polygenic correlation.

I am still concerned about the interpretation of the MR analysis for TL. The test should be for causality, not just for association. As mentioned in the response, the bidirectional MR for LOY was discarded.

The authors made a convincing argument against undertaking bidirectional MR for TL and CH.

Minor:

The Y-axis labels in Figure 2 are upside down.

Response to Reviewer Comments

We thank the Reviewers for their insightful comments during the review of our manuscript. We have drafted a revision that we believe thoroughly addresses the comments raised, which, in turn, has improved the clarity of our manuscript. We include below detailed responses to each of the Reviewers' comments and track changes in the manuscript with **highlighted text**.

Reviewers' Comments:

Reviewer #1:

Remarks to the Author:

The authors have provided an incredibly thorough response effectively addressing all my previous questions and concerns as Reviewer #1, and have added additional analyses that add extra value to a paper that was already a substantial contribution to the field. I have no further comments.

We thank Reviewer #1 for their time reviewing our manuscript and are pleased they found our response addressed all previous questions.

Reviewer #2:

The manuscript from Brown et al. is improved. The CHIP variants identified are now more in line with prior studies, which adds confidence to the results. The use of HDL also identified novel genetic associations, although most of these are expected. I remain unconvinced of the LOY association results. For example, the inverse signal in Table S5 seems to be restricted largely to DNMT3A and TET2 mutations, and is absent in other genes, which has no apparent reason or explanation. A **second** issue is that the sensitivity to detect CHIP vs mCA is very different. It is not explicitly stated here, but the MOCHA calls in UKB have a median cell fraction of ~2-4% (equivalent to VAF of 1-2%). The median VAF in the CHIP calls would probably be ~8-9%. Thus, the CHIP clones are much larger than the mCA clones, which confounds the analyses. The authors should repeat association tests for mCA to CHIP using equivalent sensitivity thresholds, i.e. restricting mCA cell fraction to $\sim >15\%$. This would impact the interpretation. For example, in the current analysis they could conclude that having a large CHIP clone blocks the growth of other clones, including mCA, due to clonal competition. However, it does not allow for any inference on mutational cooperativity since most likely the events are in different clones. Restricting to only large mCA clones would be more likely to capture events that are co-occurring in the same cells to allow for better inference on cooperativity.

We thank Reviewer #2 for their time and are pleased they found the updated analysis adds confidence to the results.

We agree with Reviewer #2 that an additional analysis is needed to better characterize non-*TET2* and non-*DNMT3A* mutations with LOY. To expand our effort, we performed an analysis that combined non-*DNMT3A* and non-*TET2* mutations and observed a negative point estimate for the LOY association that is of lesser in magnitude than the *DNMT3A* and *TET2* associations and statistically insignificant. This result provides support that *DNMT3A* and *TET2* are likely the primary CHIP mutations responsible for the overall inverse relationship observed between CHIP and LOY, whereas other less frequent CHIP mutations potentially lack this relationship; although power was limited to characterize the LOY relationship with less frequent CHIP mutations. We have revised Supplementary Table 5 to include the new non-*DNMT3A* and non-*TET2* mutation group as shown below.

Supplemental Table 5. CHIP-LOY associations by CHIP gene. Associations were adjusted for age, age-squared, and detailed smoking status and restricted to males.

CHIP-gene	N _{CHIP}	T-statistic	p-value
DNMT3A	1818	-3.71	2.08E-04
Non- DNMT3A	1500	-2.03	4.23E-02
TET2	786	-3.99	6.50E-05
Non- TET2	2532	-2.47	1.34E-02
Non-TET2 & Non-DNMT3A	783	-0.32	7.50E-01
ASXL1	347	-0.23	8.18E-01
PPM1D	102	-1.55	1.21E-01
TP53	66	0.75	4.54E-01
SRSF2	69	-2.24	2.48E-02
SF3B1	51	0.73	4.65E-01
ATM	21	0.47	6.40E-01

In addition, we have updated the text in the Results section as follows:

“The inverse association and exclusivity between CHIP and LOY were most prominent when stratified by the frequently observed CHIP gene mutations *DNMT3A* CHIP with LOY (N_{CHIP}= 1,818, T-statistic= -3.71, P= 2.08×10⁻⁴) and *TET2* CHIP with LOY (N_{CHIP}= 786, T-statistic= -3.99, P= 6.50×10⁻⁵) (**Supplemental Table 5**). Limited evidence was observed for an inverse relationship with LOY for less common CHIP mutations, suggesting *DNMT3A* and *TET2* mutations are primarily responsible for the overall inverse relationship observed between CHIP and LOY.”

With respect to the second point, we agree with Reviewer #2 that the CHIP clones we detected are much larger than the mCA clones. Following the reviewer’s suggestion, we repeated association tests between mCA and CHIP restricting to mCA cell fractions above 15%. Results indicate stronger associations between mCAs and CHIP VAF when restricting to mCAs with cell fractions above 15%. The highlighted rows below have been added to Supplemental Table 12.

Supplemental Table 12. CHIP VAF association (β) with 95% confidence intervals by CHIP and autosomal mCA status. Associations are adjusted for age, age-squared, sex, and detailed smoking status.

Mutation status	N	beta	sd	lower_ci	upper_ci	pvalue
CHIP only	6841			referent		
CHIP and Autosomal mCA	386	0.0134	0.0034	0.0066	0.0201	1.03E-04
CHIP and Autosomal mCA > 15% cell fraction	74	0.0306	0.0076	0.0156	0.0455	5.97E-05
Overlapping CHIP and Autosomal mCA	53	0.0592	0.0091	0.0414	0.0769	6.75E-11
Overlapping CHIP and Autosomal mCA > 15% cell fraction	16	0.0955	0.0163	0.0636	0.1274	4.70E-09

Supplemental Figure 10 was also updated to include the analyses restricted to higher cell fractions.

Supplemental Figure 10. CHIP VAF association (β) with 95% confidence intervals by CHIP and autosomal mCA status. The number of subjects within each group from UK Biobank are given. Associations are adjusted for age, age-squared, sex, and detailed smoking status. Detailed results are given in **Supplemental Table 12**.

We have also updated the text in the Results section as follows:

“As detection of CHIP requires higher VAF than cell fractions required to detect autosomal mCAs, we performed sensitivity analyses restricting to autosomal mCAs with cell fractions similar to the detection level for CHIP mutations (cell fraction >15%) to eliminate any potential bias due to detection differences. Results from the sensitivity analysis showed similar significant associations with higher estimated effect sizes, further supporting potential mutational

cooperativity between CHIP and autosomal mCAs (**Supplemental Figure 10, Supplemental Table 12**).’

Minor:

The supplemental tables need better annotations and consistency. E.g. why is on the T-statistic reported by Table S5, not odds ratios or confidence intervals? I have no clue what Table S8-9 are meant to be. What do the column names mean in Table S10-11?

We thank Reviewer #2 for identifying the need to clarify annotations in the Supplementary Tables. We have reviewed all Supplemental Tables and updated titles to be more descriptive and altered column names to improve clarity. With respect to Table S5, there were no odds ratios as the test was based on a linear model; this has been clarified in the table description. Tables S8-9 are intended to indicate the distribution of other CH types by clonal fraction of autosomal mCAs (Table S8) and variant allele fraction of CHIP (Table S9). The column name for tables S10-11 have been renamed to indicate the number of individuals and the proportion of individuals.

Reviewer #3:

Remarks to the Author:

In the revised paper by Brown et al., the authors elegantly refined the analyses, resulting in several additional findings. The HDL analysis revealed more correlations and generate further insights. Additional deep investigations supported the interesting inverse association between CHIP and LOY, as well as the CHIP-preceding-mCA hypothesis. The new analysis provided a comprehensive view of the combination of multiple CH types for the risk of hematological malignancies. The GWAS of MPN, empowered by shared genetic correlation of CH traits, identified 25 MPN loci, more than double the previous number.

The authors decided not to correct for multiple testing, but now included a caution message in the discussion, and discarded the term “significant” from some parts of the text, which is now appropriate I think.

We thank Reviewer #3 for the review of our revised manuscript and are pleased the new analyses and additional caution message in the Discussion were positively received.

The revised paper has resolved some conflicting results. However, the LOY-MPN association still shows opposite directions in genetic and phenotypic correlations. This reviewer believes that this may be due to the very low h^2 value of MPN sumstats, which led to an unreliable GC analysis result in this study. To support the results, if the authors intend not to remove genetic correlation studies about MPN, it might be better to perform analysis only with the effects of known significant variants for comparison, in addition to polygenic correlation.

We agree the opposite directions of the genetic and phenotypic correlations for the LOY-MPN association appear conflicting. The reported h^2 estimate in Supplementary Table 29 is low because this estimate is for observed heritability and does not account for the low population

prevalence of MPN. The liability scale heritability is approximately 6.5% (doi: [10.1038/s41586-020-2786-7](https://doi.org/10.1038/s41586-020-2786-7)). As we observe a significant genetic correlation between MPN with LOY using two different methods (HDL and LDSC), we do not believe the lower MPN heritability is leading to a spurious positive genetic correlation. The summary statistic-based pathway analysis further supports a shared genetic component as LOY and MPN such that both traits are associated with common gene sets (e.g., hematopoietic progenitor cells, hematopoietic cell lineage and differentiation gene sets, DNA damage response and telomere stress induced senescence). In support of the LOY-MPN genetic correlation, we also observed strong phenotypic associations of LOY and MPN with myeloid traits that closely mirror the magnitude and significance of the genetic correlation results. Further, we used shared genetic architecture of MPN with LOY to identify 15 genetic loci not previously identified as associated with MPN. Together, these multiple lines of evidence support a shared polygenic architecture of LOY and MPN in which the two traits have a positive genetic correlation. In an effort to be cautious, we have edited the Discussion on the LOY-MPN relationship to read as follows:

“A notable discordance in directionality is between the genetic correlation and the phenotypic relationship between LOY and MPN; although further genetic investigations in larger MPN sets are needed to replicate these findings. This potential discordance supports a shared genetic etiology as indicated by the genetic associations, but the phenotypic relationship indicates a mutual exclusivity between LOY and MPN suggesting that for some types of CH when one type of CH develops, the occurrence of others could be suppressed (e.g., DNMT3A and TET2 CHIP³⁹).”

I am still concerned about the interpretation of the MR analysis for TL. The test should be for causality, not just for association. As mentioned in the response, the bidirectional MR for LOY was discarded.

We agree with Reviewer #3 and have corrected the wording in the TL MR section to reflect we are testing for causality and not just for an association. The relevant section now reads:

“To further test for a causal relationship with TL, we conducted one-direction Mendelian randomization (MR) between TL and each CH type using 130 previously published TL-associated variants (Supplemental Figure 13).²⁷ Based on MR-IVW models, we observed a positive relationship between the TL IV and autosomal mCAs ($Z_{\text{filtered}} = 5.65$, $P = 1.21 \times 10^{-7}$), CHIP ($Z_{\text{filtered}} = 5.72$, $P = 9.65 \times 10^{-8}$), and MPNs ($Z_{\text{filtered}} = 5.61$, $P = 1.88 \times 10^{-7}$), and observed a negative relationship between the TL IV and LOY ($Z_{\text{filtered}} = -6.40$, $P = 8.11 \times 10^{-9}$) and did not identify evidence for a causal relationship between telomere length and LOX (Supplemental Figure 14, Supplemental Table 19). These observations provide additional support of a causal relationship between inherited telomere length and select CH traits.^{12,22,28–31}”

The authors made a convincing argument against undertaking bidirectional MR for TL and CH.

We agree the poor instruments for CHIP, autosomal mCAs, and LOX did not produce convincing evidence for interpretation and should not be included in the manuscript.

The Y-axis labels in Figure 2 are upside down.

Thanks for noticing this oddity. It appears the figure was rotated during the submission process when generating the merged PDF file. The submitted figure, as pasted below, is correctly oriented. We will confirm the figure is correctly oriented in any future proofs resulting from this manuscript.

Minor:

REVIEWERS' COMMENTS

Reviewer #2 (Remarks to the Author):

I thank the authors for addressing my comments. The inclusion of the 15% size threshold has improved the analysis in Table S12. However, the authors should also report the results for the analysis Table S5, the linear association of LOY with different CHIP drivers, using this threshold.

Response to Reviewer Comments

We thank the Reviewer for their insightful comment during the final review of our manuscript. Please see the detailed response below to the final reviewer comment. Table S5 was updated and highlighted text in the manuscript detail the changes made.

Reviewers' Comments:

Reviewer #2 (Remarks to the Author):

I thank the authors for addressing my comments. The inclusion of the 15% size threshold has improved the analysis in Table S12. However, the authors should also report the results for the analysis Table S5, the linear association of LOY with different CHIP drivers, using this threshold.

We thank Reviewer #2 for their time reviewing our manuscript and are pleased they found our prior responses addressed their concerns. We have added the >15% cell fraction threshold to Table S5 as requested (see below) and revised the manuscript text highlighting this sensitivity analysis conducted among higher cell fraction LOY events.

Supplemental Table 5. Linear associations between LOY and CHIP by CHIP gene. Associations were adjusted for age, age-squared, and detailed smoking status and restricted to males.

CHIP-gene	Overall LOY			LOY > 15% cell fraction		
	N _{CHIP}	T-statistic	p-value	N _{CHIP}	T-statistic	p-value
DNMT3A	1818	-3.71	2.08E-04	1467	-2.18	2.91E-02
Non- DNMT3A	1500	-2.03	4.23E-02	1221	4.20	2.74E-05
TET2	786	-3.99	6.50E-05	662	-1.71	8.71E-02
Non- TET2	2532	-2.47	1.34E-02	2026	-1.53	1.25E-01
Non- TET2 & Non- DNMT3A	783	-0.32	7.50E-01	624	2.64	8.18E-03
ASXL1	347	-0.23	8.18E-01	270	-0.70	4.82E-01
PPM1D	102	-1.55	1.21E-01	82	-0.20	9.87E-01
TP53	66	0.75	4.54E-01	57	-0.57	5.72E-01
SRSF2	69	-2.24	2.48E-02	54	-1.09	2.81E-02
SF3B1	51	0.73	4.65E-01	40	-0.34	7.36E-01
ATM	21	0.47	6.40E-01	19	-0.16	8.76E-01